# Physical processes of cooling and megadrought in 4.2 ka BP event: results from TraCE-21ka simulations

Mi Yan[1,2], Jian Liu[1,2*]

[1]Key Laboratory of Virtual Geographic Environment, Ministry of Education; State

key Laboratory of Geographical Environment Evolution, Jiangsu Provincial

Cultivation Base; School of Geographical Science, Nanjing Normal University,

Nanjing, 210023, China

[2]Jiangsu Center for Collaborative Innovation in Geographical Information Resource

Development and Application, Nanjing, 210023, China

[*]jliu@njnu.edu.cn

Abstract
It is widely believed that multidecadal to centennial cooling and drought occurred from
4500 BP to 3900 BP, known as the 4.2 ka BP event that triggered the collapse of several
cultures. However, whether this event was a global event or a regional event and what
caused this event remain unclear. In this study, we investigated the spatiotemporal
characteristics, the possible causes and the related physical processes of the event using
a set of long-term climate simulations, including one all-forcing experiment and four
single-forcing experiments. The results derived from the all-forcing experiment show
that this event occurs over most parts of the Northern Hemisphere (NH), indicating that
this event could have been a hemispheric event. The cooler NH and warmer Southern
Hemisphere (SH) illustrate that this event could be related to the slowdown of the
Atlantic Meridional Overturning Circulation (AMOC). The comparison between the
all-forcing experiment and the single-forcing experiments indicates that this event
might be caused by internal variability, while external forcings such as orbital and
greenhouse gases might have modulation effects. A positive North Atlantic Oscillation
(NAO)-like pattern in the atmosphere (low troposphere) triggered a negative Atlantic
Multidecadal Oscillation (AMO)-like pattern in the ocean, which then triggered a
Circumglobal Teleconnection (CGT)-like wave train pattern in the atmosphere (high
troposphere). The positive NAO-like pattern and the CGT-like pattern are the direct
physical processes that lead to the NH cooling and megadrought. The AMO-like pattern
plays a "bridge" role in maintaining this barotropic structure in the atmosphere at a
multidecadal-centennial time scale. Our work provides a global image and dynamic
background to help better understand the 4.2 ka BP event.



## 1 Introduction

Understanding the characteristics and mechanisms of climate changes during the Holocene can help predicting future changes. The multidecadal-to-centennial abrupt climate change, or the rapid climatic change during ca. 4.5-3.9 ka BP (before 1950 CE), the so called "4.2 ka BP event", was one of the major climate events during the Holocene (Wang, 2009; Staubwasser and Weiss, 2006; Mayewski et al., 2004; Wang, 2010). This event is considered to be closely linked to the cultural evolutions of different regions of Eurasia such as the collapse of the Akkadian empire, the termination of the urban Harappan civilization in the Indus valley and the collapse of Neolithic Cultures around the Central Plain of China (Weiss et al., 1993; Weiss and Bradley, 2001; Wu and Liu, 2001; Staubwasser et al., 2003; Wu and Liu, 2004; An et al., 2005; Staubwasser and Weiss, 2006; Liu et al., 2013; Weiss, 2015, 2016). Moreover, this event is also thought to be the transition of the Middle to Late Holocene (Walker et al., 2012; Finkenbinder et al., 2016). However, the characteristics, causes and corresponding mechanisms behind this event remain unclear.

The 4.2 ka BP event is mostly characterized by rapid events at various latitudes (Jansen et al., 2007), e.g., cooling in Europe (Lauritzen, 2003), centennial megadroughts in North America (Booth et al., 2005), decreased precipitation in both southern and northern China (Tan et al., 2008), and the weakened summer monsoon in India (Nakamura et al., 2016); however, the manifestation of this event is far from convincing and needs more evidence and simulation investigations (Roland et al., 2014). Many reconstructions have shown that the 4.2 ka BP event is dominated by megadroughts at centennial-scale over mid-low latitudes (Tan et al., 2008; Yang et al., 2015; Weiss, 2016). However, Roland et al. (2014) found no compelling evidence, at least in peatland records, to support that there was a 4.2 ka BP event in Great Britain and Ireland. Moreover, according to the hydrologic cycle (i.e. the hydroclimate changes are often regionally specific), it cannot be ruled out that there were no flooding events somewhere else during this period. For example, Huang et al. (2011) and Tan et al. (2018) found that successive floods occurred over the middle reaches of the Yellow River in China in association with the abrupt climatic event of 4.2 ka BP.

Understanding the causes and mechanisms of the 4.2 ka BP event can provide
explanations for the reconstructed discrepancies over different regions. For the causes
of the event, some reconstruction and modeling studies have suggested that the solar
irradiance could have played an important role in the early Holocene climate changes
(Wang et al., 2005; Rupper et al., 2009; Owen and Dortch, 2014); however, no strong
evidence has shown that the solar irradiance affected glacier fluctuations (cooling
events) in the late Holocene since there is yet no good mechanistic explanations of how
small changes in solar irradiance could significantly affect large scale climate changes
(Solomina et al., 2015). Tan et al. (2008) thought that the 4.2 ka BP event could have
been induced by the southward shift of the Intertropical Convergence Zone (ITCZ) and
oceanic sea surface temperature (SST) changes, as well as the vegetation feedback
caused by the solar activity. Liu et al. (2013) and Deininger et al. (2017) argued that the
atmospheric circulation, such as the North Atlantic Oscillation (NAO)-like pattern but
on a centennial time scale, could have played a more important role than the ocean
circulation in this event, although the mechanisms that forced the circulation change
remained unclear. A new reconstruction study has also shown that the dry phases over
the western Mediterranean in the period of 4.5 ka BP-2.8 ka BP generally agreed with
positive NAO conditions (Ramos-Román et al., 2018). However, studies come to
different conclusions on the likely phase of the NAO-like patter during the late
Holocene (Finkenbinder et al., 2016). Some studies show positive NAO-type patterns
during the late Holocene (Tremblay et al., 1997; Sachs, 2007; Ramos-Román et al.,
2018), whereas others show negative NAO-like patterns (Rimbu et al., 2004). Since the
mechanisms could be a complex set of air-sea interactions (Roland et al., 2014), it is
hard for reconstruction to provide a general record due to its limitations such as
interpretation and spatially incompleteness. The mechanisms behind the 4.2 ka BP
event need to be clarified.
Therefore, to improve understanding of the 4.2 ka BP event, new high-resolution
reconstruction studies that focus on the 4.2 ka BP event are required. On the other hand,
physical-based modeling research can provide general concepts of the characteristics
of the event along with the causes and the mechanisms. Climate simulations have been

conducted to investigate another abrupt cooling event in the early Holocene, the so-called 8.2 ka BP event. The simulations were used to test the hypothesis raised by the reconstruction studies that the 8.2 ka BP event was most likely caused by freshwater forcing and was associated with weakening of the Atlantic Meridional Overturning Circulation (AMOC) (Morrill et al., 2013; Wagner et al., 2013; Morrill et al., 2014; Matero et al., 2017; Ljung et al., 2008; Alley and Agustsdottir, 2005). For example, the simulations argued that the meltwater from the collapse of the ice dome over Hudson Bay was an essential forcing of the 8.2 ka BP event (Wagner et al., 2013; Matero et al., 2017). However, little modeling work has been applied to the 4.2 ka BP event.

Recently, Ning et al. (2019) briefly compared the spatial patterns of climate change in the 9[th] and 5[th] millennia BP using a set of transient modeling results on a long-term perspective. In the present study, we will use the same set of simulation results to provide an in-depth characteristics of the 4.2 ka BP event and will focus on the possible causes and mechanisms behind this event. The model and experiments are introduced in Sect. 2. The results are shown in Sect. 3. The possible causes and mechanisms are discussed in Sect. 4, and conclusions are drawn in Sect. 5.

## 2 Model and experiments

A set of transient simulations (TraCE-21ka, Simulation of Transient Climate Evolution over the past 21,000 years, He, 2011) conducted with the Community Climate System model version 3 (CCSM3) was used to investigate the spatial and temporal characteristics of the 4.2 ka BP event and to determine the possible causes and mechanisms behind this event. The experiments are listed in Table 1, including one transient experiment with all-forcings (TraCE-ALL), one single-forcing experiment forced only by transient orbital variation (TraCE-ORB), one single-forcing experiment forced only by transient melt-water flux (TraCE-MWF), one single-forcing experiment forced only by quasi-transient ice-sheet (TraCE-ICE), and one single-forcing experiment forced only by transient greenhouse gases concentrations changes (TraCE-GHG). The simulations were conducted from 22000 BP to 1990 CE for the TraCE-ALL, the TraCE-ORB and the TraCE-GHG experiments, and from 19000 BP to 1990 CE for

the TraCE-MWF and the TraCE-ICE experiments.

The transient June insolation changes at 60°N and 60°S that resulted from the

orbital variation and the transient $CO_2$ change used in the simulations are shown in Fig.
1. The continental ice-sheet and topography changes are based on the ICE-5G (VM2)
reconstruction (He et al., 2013; Peltier, 2004). For the geography changes, the Barents
Sea opens at 13.1 ka BP, the Bering Strait opens at 12.9 ka BP, Hudson Bay opens at
7.6 ka BP, and the Indonesian Throughflow opens at 6.2 ka BP. The freshwater injected
into Northern Hemisphere (NH) and Southern Hemisphere (SH) oceans are based on
specific time slices (e.g., 19 ka BP into North Atlantic, 17 ka BP into North Atlantic,
11.5 ka BP into Arctic, St. Lawrence River, Hudson Strait, Barents Sea, North Sea, Ross
Sea and Weddell Sea). Note that no freshwater was delivered to the ocean after 5000
BP in the TraCE-ALL and TraCE-MWF experiments. The detailed information about
the experiments design can be referred to He (2011) and He et al. (2013).

The TraCE-21ka simulation was evaluated with reconstructions and was found

that it could reproduce major deglacial temperature evolutions (Clark et al., 2012;
Shakun et al., 2012). It has been used to depict the causes and mechanisms of Holocene
climate changes, such as the Bølling-Allerød warming (Liu et al., 2009), cooling into
the Younger Dryas and recovery to warm conditions (Liu et al., 2012) and the ENSO
evolution over the past 21 ka (Liu et al., 2014a). In the present work, we adopted the
period of 5000 BP-3000 BP to focus on the 4.2 ka BP event.

**3 Results**
3.1 Identification of 4.2 ka BP event in the model simulation

The 101-year running mean annual NH surface temperature and precipitation

during 5 ka BP-3 ka BP shows double peak centennial cooling and drought from 4.4 ka
BP to 4.0 ka BP (Fig. 2, dashed black line). However, the variabilities are smaller over
the SH than those over the NH. There is no significant cooling and drought event during
that period (Fig. S1, dashed black line) over the SH. The SH precipitation even shows
a double-peak wet condition during the period of 4.4 ka BP-4.0 ka BP.

The double peak centennial cooling and drought are still obvious when the 31-year

running mean is applied to the time series (not shown), which indicates that the
simulated climate events potentially comparable to the 4.2 ka event. Moreover, the
centennial warming periods right before and after the cooling event indicate that this
event might be included in a quasi-millennium variation. Therefore, the 4.2 ka BP event
could be a multiscale event, i.e. from multi-decadal to millennium.

The seasonal mean NH surface temperature changes show that the annual mean

variability is dominated by the boreal winter (December-January-February, DJF)
surface temperature change (Fig. S2). The correlation coefficient between the annual
mean NH surface temperature (NHT) and the DJF mean NHT is 0.96 (after the 101-
year running mean), which is significant above the 99% confidence level, much higher
than the correlation coefficient between the annual mean and the boreal summer (June-
July-August, JJA) mean of only 0.30 (after the 101-year running mean), which is not
significant. However, this is different for the precipitation change, for which both the
JJA mean and the DJF mean contribute to the annual mean precipitation change (not
shown).

To identify the characteristics of the 4.2 ka BP event, two centennial cool periods

and two centennial warm periods that exceeded ±0.5 standard deviations are selected.
The two centennial cool periods span from 4320 BP to 4220 BP and from 4150 BP to
4050 BP, and the two centennial warm periods span from 4710 BP to 4610 BP and from
3980 BP to 3880 BP.

3.2 Spatial characteristics of surface temperature and precipitation

To help draw a coherent global view of the 4.2 ka BP event, the spatial

characteristics of temperature and precipitation changes during the 4.2 ka BP event are
shown in Fig. 3.

Figure 3a gives the spatial distribution of the annual mean surface temperature

difference between the cold periods and the warm periods. The cooling significantly
occurred over most regions of the NH, especially over the middle to high latitudes of
the NH and most land regions of the SH. Most parts of India, northern Mexico and the
middle latitudes of the SH ocean experienced warm conditions. Such asymmetric
change between the hemispheres (cool NH and warm SH) favors the southward shift of
the ITCZ. The spatial distribution of the surface temperature change is still dominated
by the boreal winter pattern (not shown). The large cooling over the NH and small
warming over the SH could be related to the orbital change (Fig. S3), which induces
insolation increasing over the SH but decreasing over the NH.

The spatial distribution of annual mean precipitation differences between the cold

periods and the warm periods is shown in Fig. 3b. During the cold periods, significant
drought is mainly located over many land regions of the NH, especially over Europe,
western Asia, and interior North America and Central America. The significant dry
conditions over the Dead Sea, the Gulf of Omen, interior North America and western
North Africa and the wet condition over South America are consistent with the
reconstructions (Yechieli et al., 1993; Cullen et al., 2000; Forman et al., 1995; Marchant
and Hooghiemstra, 2004). For the SH, the land precipitation increased, which indicates a
southward shift of the ITCZ, as suggested by the aforementioned asymmetric
temperature change and by the previous studies based on both reconstructions
(Fleitmann et al., 2007; Cai et al., 2012) and simulations (Broccoli et al., 2006). Over
East China, the precipitation anomalies show a wet south-dry north pattern, which
indicates a weakened East Asian monsoon revealed by the reconstruction record (Tan
et al., 2018). However, the simulated anomaly pattern is not very significant over East
China. This might be related to the model resolution, the model performance, or the
actual climate change. Therefore, simulations with higher resolution, inter-model and
model-data comparisons are required to draw a clearer view about the climate change
over East China.

The sea surface temperature (SST) shows that the largest change occurs over the

northern Atlantic Ocean and then the northern Pacific Ocean (Fig. 4). The warmer south
and cooler north over the Atlantic Ocean indicates an Atlantic Multi-Decadal
Oscillation (AMO)-like pattern with its cold phase. The cold phase of the AMO has
been confirmed to induce summer rainfall decreases over India and Sahel in both
simulations and proxy data (Zhang and Delworth, 2006; Shanahan et al., 2009).

The simulated characteristics of the temperature change, the precipitation change,

and the SST change are similar to those responses to the weakened AMOC state
(Vellinga and Wood, 2002; Zhang and Delworth, 2005; Delworth and Zeng, 2012;
Brown and Galbraith, 2016) (Fig. S4).

3.3 Circulations associate with the 4.2 ka BP event

The sea level pressure (SLP) differences between the cooler periods and the

warmer periods show that the largest change occurs over the mid-high latitudes of the
NH and SH (Fig. 5a). The negative SLP anomalies over the high North Atlantic and
positive SLP anomalies over the middle North Atlantic during the cool periods resemble
a positive North Atlantic Oscillation (NAO)-like pattern but on a centennial-millennial
time scale. The positive NAO-like pattern is accompanied by cyclonic circulation over
Iceland and anticyclonic circulation over the Azores Islands and thus strengthened
westerlies over the downstream regions (Fig. 5a). The subtropical highs and the relative
anticyclones in both the SH and NH are strengthened during the cold periods from low
troposphere (850 hPa) to high troposphere (200 hPa), which illustrates a barotropic
structure (Fig. 5). Note that the strengthened subtropical highs over the NH are most
significant at low level (sea level and 850 hPa), while the subtropical highs over the SH
are most significant at high level (200 hPa). The centers with positive geopotential
height anomalies during the 4.2 ka BP event over Western Europe, Central Asia, East
Asia, the east north Pacific and Eastern North America, as well as the anti-cyclonic
circulation anomalies at 200 hPa (Fig. 5d), resemble a Circumglobal Teleconnection
(CGT)-like wave pattern (Ding and Wang, 2005; Lin et al., 2016) but on a centennial-
millennial time scale.

The strengthened subtropical highs with mid-latitudes anticyclones from lower to

upper levels are the direct physical processes that cause the precipitation decreases and
thus the following megadrought over mid-latitudes of NH regions, particularly over
Eurasia. The cooler land-warmer ocean over East Asia and the West Pacific (Fig. 3a)
indicate weakened land-ocean thermal contrast associated with significantly higher SLP
over land and lower SLP over the adjacent ocean (insignificant) (Fig. 5a). The
weakened land-ocean contrast can lead to a weaker East Asian monsoon, accompanied

by precipitation increases over the southern China pattern and precipitation decreases over the northern China pattern (Fig. 3b). Such conclusion is very rough, since the simulated anomaly patterns are not very significant. More investigations with higher resolutions of modeling and reconstruction works are required to get satisfactory results.

**4 Discussions**

The simulations show that the cool and dry conditions of the 4.2 ka BP event is more like a hemispheric phenomenon, mainly located over the NH, rather than a global phenomenon. The land over the SH experiences cool but wet conditions, and the mid-latitude SH ocean is warmer. The potential causes and mechanisms of this event will be discussed in this section.

4.1 The possible causes of the 4.2 ka BP event

Some records suggested that solar irradiance was one of the essential mechanisms that drove the Holocene climate variation at centennial to millennial time scales (Bond et al., 2001), whereas others suggested that the linkage between solar irradiance and multicentury scale cooling events during the Holocene was weak, particularly in the mid- to late-Holocene (Turney et al., 2005; Wanner et al., 2008). Changes in solar irradiance are not included in the experiments used in the present work. Nonetheless, we still obtain multicentury cooling events (such as the 4.2 ka BP event) in the TraCE-ALL experiment, but with smaller magnitude. This side-fact indicates that the solar irradiance might not be the driving factor for the Holocene cooling events.

If the results derived from the TraCE-ALL experiment are consistent with those derived from a particular single-forcing sensitivity experiment, we assume the variation to be forced by that forcing. Otherwise, if the results derived from the TraCE-ALL experiment differ from those from the single-forcing sensitivity experiments, we assume the variation to be forced by the internal variability. In this section, we use the series after applications of 101-year running means as an example and compare the results derived from the all-forcing experiment to those derived from the single-forcing experiment to determine the possible forcings that triggered the 4.2 ka BP event.

The correlation coefficients between the annual mean NHT derived from the

TraCE-ALL run and the NHT derived from each single-forcing run are listed in Table
2. A two-sided Students t-test is used for the statistical significant test, assuming 20
degrees of freedom, which is estimated simply from a 2000-year time series subjected
to a 100-year running mean (Delworth and Zeng, 2012). There is no significant clue
that the annual mean NHT variation is forced by the orbital variation or the other
forcings due to the non-significant correlations. During the period of 5000 BP - 3000
BP, the variation of simulated JJA mean NHT is likely forced by the solar radiation due
to the orbital variation (Table 2; the correlation coefficient between the two series is
0.79 at $p<0.05$), whereas the greenhouse gas change has a comparable negative impact
on the JJA mean NHT (the correlation coefficient is -0.73 at $p<0.05$). The melt-water
flux also has a moderate contribution to the JJA mean NHT change (the correlation
coefficient is 0.48 at $p<0.05$). For the DJF mean NHT, only melt-water flux has a
notable negative effect (the correlation coefficient is -0.43 at $p<0.05$). However, there
is no meltwater forcing during this period, so the NHT change can be taken as internal
variability. Therefore, the significant correlation coefficient between the all forcing run
result and the meltwater forcing run result might be a coincidence, due to the
autocorrelation of internal variability. This is another side-fact indicating the cold
events during the late Holocene might be related to the internal variability. Note that if
the effective degree of freedom is used, none of the abovementioned correlation
coefficients are significant. The effective degree of freedom is calculated by the
following equation:
$$N_{dof} = N \times \frac{1 - r1 \times r2}{1 + r1 \times r2}$$
where $N_{dof}$ is the effective degree of freedom regarding to the two correlation samples,
N is the total sample size, $r_1$ and $r_2$ are autocorrelation lag-1 values for sample 1 and
sample 2, respectively (Bretherton et al., 1999).
On the other hand, the annual mean NHT difference between the TraCE-ALL run
and the sum of the 4 single-forcing sensitivity experiments shows variation similar to
the NHT derived from the TraCE-ALL run from 5000 BP to 3000 BP (Fig. S5). The
correlation coefficient between these two time-series is 0.66, which is significant above

the 95% confidence level (assuming 20 degrees of freedom). We assume the difference between the TraCE-ALL run and the sum of the 4 single forcing runs to be the internal variation, taking that the climate responses to the external forcings are linear at global and hemispheric scales. Therefore, the internal variation might play a dominant role in the climatic variation during the period of 5000 BP-3000 BP. However, the linearity of the climate responding to the external forcings need further clarification, since there would be interactions between each forcing and between forcings and internal variability.

Moreover, there is no double-peak cooling event during the period of 4400 BP-4000 BP in any single forcing run (Fig. 1, colored lines), which indicates that the 4.2 ka BP event might not be triggered by those external forcings, including the orbital, the melt-water flux, the ice-sheets and the greenhouse gases in isolation. Volcanic eruptions have been identified as one of the important drivers of climate variation, whereas there were few eruptions during 4400 BP-4000 BP (Sigl et al., 2018). Therefore, we conclude that the variability relating to the 4.2 ka BP event might be driven by the internal variability. Klus et al. (2017) also suggested that the internal climate variability could trigger abrupt cold events in the North Atlantic without external forcings (e.g., solar irradiance or volcanic).

However, why such large variation due to the internal variability occurs at approximately 4.2 ka BP remains unknown. There is little ice-sheet change and no melt water discharge after 5.0 ka BP in the TraCE-ICE run and TraCE-MWF run, and the variations of climate derived from these two runs can thus be considered as internal variabilities. The multicentennial cooling events can also be found in the standardized NHT during the last 5000 years of the two experiments (Fig. S6), and there are drought events in the standardized NH precipitation time series (not shown). However, the timing of those cooling and drought events occurs stochastically. This indicates a general concept of the random variation of the internal mode of the climate system. There is a reduction of NH temperature and precipitation at around 4600 BP in the TraCE-ORB (Fig. 2, orange lines), which might be related to the timing of the event as speculated by Ning et al. (2019).

Ning et al. (2019) compared the 5[th] millennium BP cooling with the 9[th] millennium
cooling and concluded that the 9[th] millennium BP cooling was resulted from the
freshwater forcing while the orbital forcing is the most likely explanation of cooling in
the North Atlantic starting from the early 5[th] Millennium BP through most of the later
Holocene, but with fluctuations. In the present work, we attribute this fluctuation to the
internal variability, which is superposed on the orbital induced long-term trend. This
work and Ning et al.'s work (2019) focus on different aspects and different time scales,
and are complementary to better understand the 4.2 ka BP event.

4.2 The mechanisms of the centennial-millennial cooling and drought
As has mentioned in Sec. 3.3, the low level NAO-like pattern and upper level
CGT-like pattern are the direct mechanisms that cause cooling and megadroughts over
most part of the NH. Previous studies also proposed that the temperature and
precipitation changes over Eurasia and Africa were directly linked to the NAO (Cullen
et al., 2002; Kushnir and Stein, 2010). The first leading mode of the Empirical
Orthogonal Function (EOF) of the annual mean SLP during 5 ka BP-3 ka BP shows a
double-peak positive NAO-like pattern but on a centennial scale during the period of
4400 BP-4000 BP (Fig. 6). The first leading EOF of geopotential height at 200 hPa after
application of a 31-year running mean shows a CGT-like pattern and similar double-
peak variation during the period of 4400 BP-4000 BP, which is more obvious after
applying the 101-year running mean (Fig. 7). This means that the double-peak cooling
and drought of the 4.2 ka BP event could be strongly related to the double peak positive
NAO-like pattern (at low level) and CGT-like pattern (at high level) at a centennial time
scale.
Li et al. (2013) suggested that the NAO is a predictor of NHT multidecadal
variability during the 20[th] century. In this study, significant correlation is also found
between the annual mean NAO index and the annual mean NHT during the period of
4400 BP-4000 BP, with the NAO leading by approximately 40 years (Fig. 8). The NAO
index is defined by the first leading mode of the EOF of the SLP. The regressed annual
mean surface temperature against the NAO index 40 years earlier during 4400 BP and

4000 BP shows cooler NH high latitudes and a warmer SH (Fig. S7), especially the cooling over the northern North Atlantic Ocean, Europe, East Asia and North America.

The geopotential height at 200 hPa regressed against the SST over the two North Atlantic outstanding regions (Fig. 4) shows a CGT-like pattern after application of a 31-year running mean (Fig. 9), which is similar to the conclusion from Lin et al. (2016) that the CGT could be excited by the AMO-related SST anomaly. The regressed 200 hPa geopotential height shows a similar pattern after application of a 101-year running mean (not shown). The anticyclones associated with CGT-like pattern over the West Europe, Central Asia and North America can suppress the precipitation and thus lead to megadrought over these regions.

Considering the NAO-like pattern, the CGT-like pattern and the AMO-like pattern together, we suggest that the AMO could be playing a "bridge" role to keep the barotropic structure at the centennial scale, which is similar to the synthesis proposed by Li et al. (2013) that the AMO is a "bridge" that links the NAO and NHT at a multidecadal timescale. Delworth and Zeng (2016) suggested that the NAO variation had significant impact on the AMOC and the subsequent influence on the atmosphere and large-scale climate at multidecadal-centennial time scales. Other studies also focused on the role of SST anomalies over the North Pacific and North Atlantic oceans when investigating the possible mechanisms of the 4.2 ka BP event (Kim et al., 2004; Marchant and Hooghiemstra, 2004; Booth et al., 2005).

We notice the centennial-millennial variation of the AMOC after the mid-Holocene in the all forcing run (Fig. S4a). There also exits a double peak variation during the period of 4400-4000, accompanied by the similar spatial patterns of temperature and precipitation anomalies as the simulated 4.2 ka BP event (Fig. S4b, c). However, whether this AMOC variation is related to the external forcing, such as the orbital forcing, or just the internal variability remains unknown, and needs further investigations.

**5 Conclusion**

The characteristics of the 4.2 ka BP event along with the potential drivers and the

mechanisms are investigated using a set of transient climate simulations. The simulated event is characterized by hemispheric cooling and megadrought over the NH, whereas the SH experiences warming (over mid-latitude ocean) and wet conditions during this event. The annual mean temperature change is dominated by the boreal winter change. The cool and dry NH and warm and wet SH pattern indicates a southward shift of the ITCZ, as suggested by the reconstructions. These characteristics could also be related to a weakening of the AMOC, which needs further investigation.

By comparison between the all-forcing experiment and the single-forcing sensitivity experiments, the 4.2 ka BP event can largely be attributed to the internal variability, although the orbital forcing and the greenhouse gases could impact the boreal summer NHT variation. The origin could be in polar regions and the North Atlantic and may influence the NH climate through teleconnections such as the NAO-like pattern and the CGT-like pattern. The positive NAO-like pattern in the atmosphere triggers cooling over the NH and the negative AMO-like pattern in the ocean, which may last for decades or even centuries. The negative AMO-like pattern triggers CGT-like wave patterns at a multidecadal-centennial time scale accompanied by anticyclones over West Europe, Central Asia and North America, which induce megadrought over those regions. The simplified diagram of the mechanism is shown in Fig. 10.

Our findings provide a global pattern and mechanical background of the 4.2 ka BP event that can help better understanding this event. We attribute the internal variabilities to be an essential forcing of the 4.2 ka BP event. However, whether or not the external forcings have modulation effects need to be clarified. For example, is the timing of the event stochastic due to the internal variability or modulated by the external forcings such as the orbital changes? Why the SST forcing in the North Atlantic can be maintained at a multidecadal-centennial time scale requires more study. Current results are mainly based on annual mean precipitation and temperature, whereas the impacts of external forcings may have seasonal dependence; further investigations are required to evaluate these impacts.

The model responses to the external forcings are small, especially in the Holocene because of the absence of a significant change of the AMOC and the meltwater forcing

after 6 ka (Liu et al., 2014b). So we use the amplified anomalies between the cold and
warm periods, rather than simply the cold anomalies against the long-term average, to
illustrate the mechanisms of the event. We need to keep in mind that we still might not
be modeling the events comparable to the 4.2 ka BP event, particularly during the late
Holocene. More model-data, inter-model and inter-events comparisons are required to
better understand the cold events during the Holocene.


**Acknowledgments**


We acknowledge Prof. Bin Wang and two anonymous referees for the
comments helping to clarify and improve the paper. This research was jointly
supported by the National Key Research and Development Program of China (grant
no. 2016YFA0600401), the National Basic Research Program (grant no.
2015CB953804), the National Natural Science Foundation of China (grant nos.
41671197, 41420104002 and 41631175), Open Funds of State Key Laboratory of
Loess and Quaternary Geology, Institute of Earth Environment, CAS
(SKLLQG1820), and the Priority Academic Development Program of Jiangsu Higher
Education Institutions (PAPD, grant no. 164320H116). TraCE-21ka was made
possible by the DOE INCITE computing program, and supported by NCAR, the
NSFP2C2 program,and the DOE Abrupt Change and EaSM programs.

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

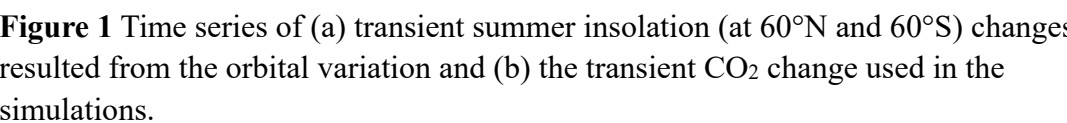

## (a) orbital-induced summer insolation

W/m2 (ref 475)

— 60N  — 60S

time (BP)

## (b) atmospheric CO2 concentration

ppmv

time (BP)

**Figure 1** Time series of (a) transient summer insolation (at 60°N and 60°S) changes
resulted from the orbital variation and (b) the transient $CO_2$ change used in the
simulations.


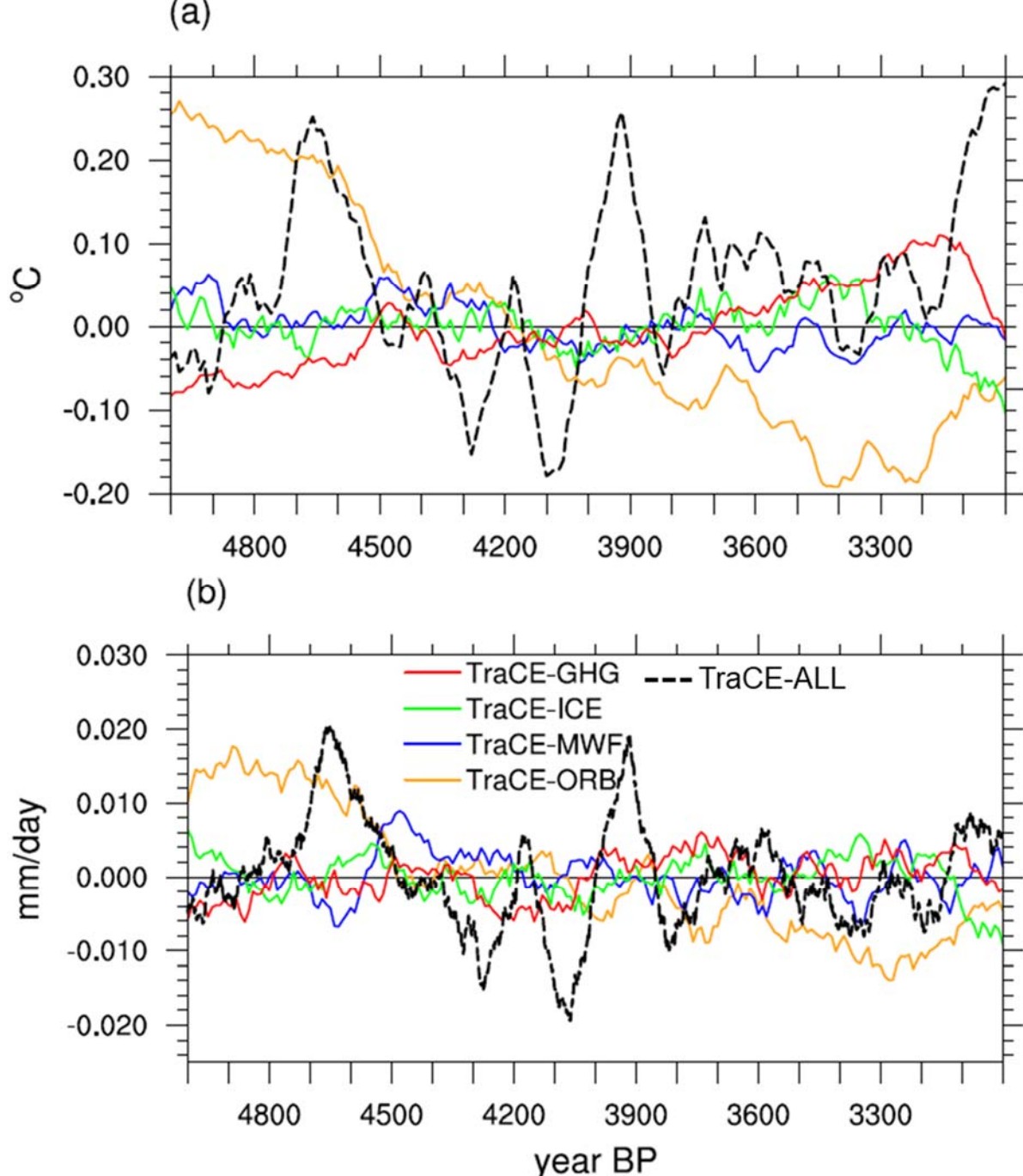


**Figure 2** Time series of annual mean NH (a) surface temperature anomalies and (b)

precipitation anomalies derived from the TraCE-ALL run (dashed black lines) and

each single forcing runs (solid color lines) from 5 ka BP to 3 ka BP. A 101-year

running mean has been applied to the time series.





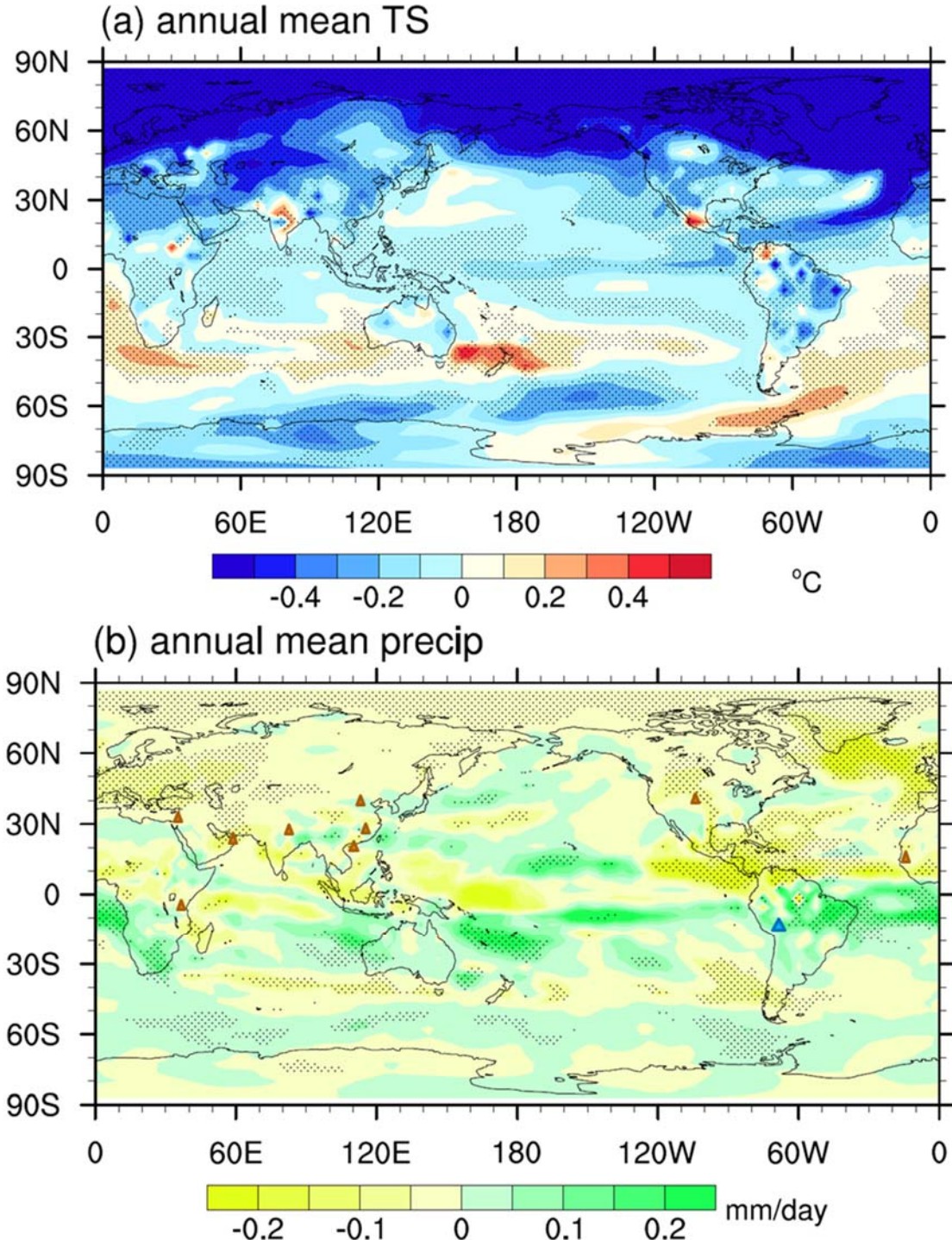

**Figure 3** Spatial distribution of the annual mean (a) surface temperature and (b)
precipitation differences between the cold periods and warm periods derived from the
TraCE-ALL run. Those regions where significant above 95% confidence level are
dotted. Triangles in (b) denote the dry (orange) and wet (blue) conditions documented

in the records, including the following sites: Kilimanjaro (3°04.6'S, 37°21.2'E)
(Thompson et al., 2002), Dead Sea (Yechieli et al., 1993), Gulf of Omen (24°23.4'N,
59°2.5'E) (Cullen et al., 2000), Lake Rara (29°32'N, 82°05'E) (Nakamura et al.,
2016), Maar lake in Huguangyan (21.15°N, 110.29°E) (Liu et al., 2000), Daihai Lake
(40.58°N, 112.7°E) (Peng et al., 2005), Poyang Lake (29.15°N, 116.27°E) (Ma et al.,
2004), Eastern Colorado Dunes (40°20'N, 104°16'E) (Forman et al., 1995), Lake
Titicaca (12.08°S, 69.85°W) and Lake Guiers (16.3°N, 16.5°W) (Marchant and
Hooghiemstra, 2004).


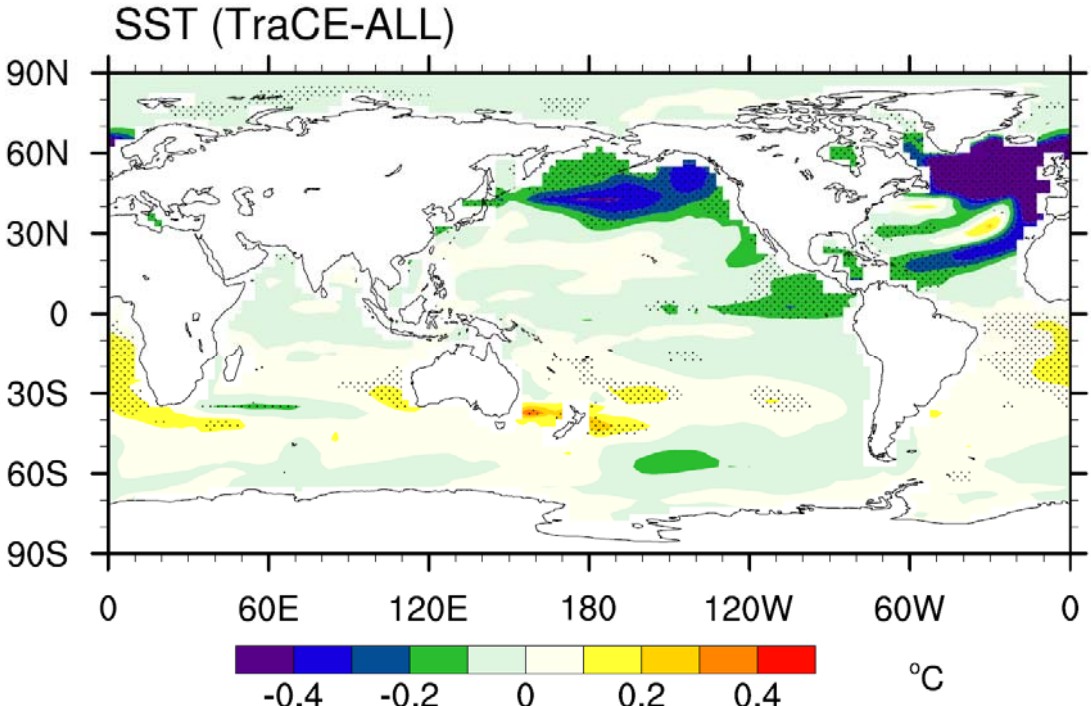


**Figure 4** Spatial distribution of annual mean SST difference between the cold and
warm periods derived from the TraCE-ALL run. Those regions where significant
above 95% confidence level are dotted.




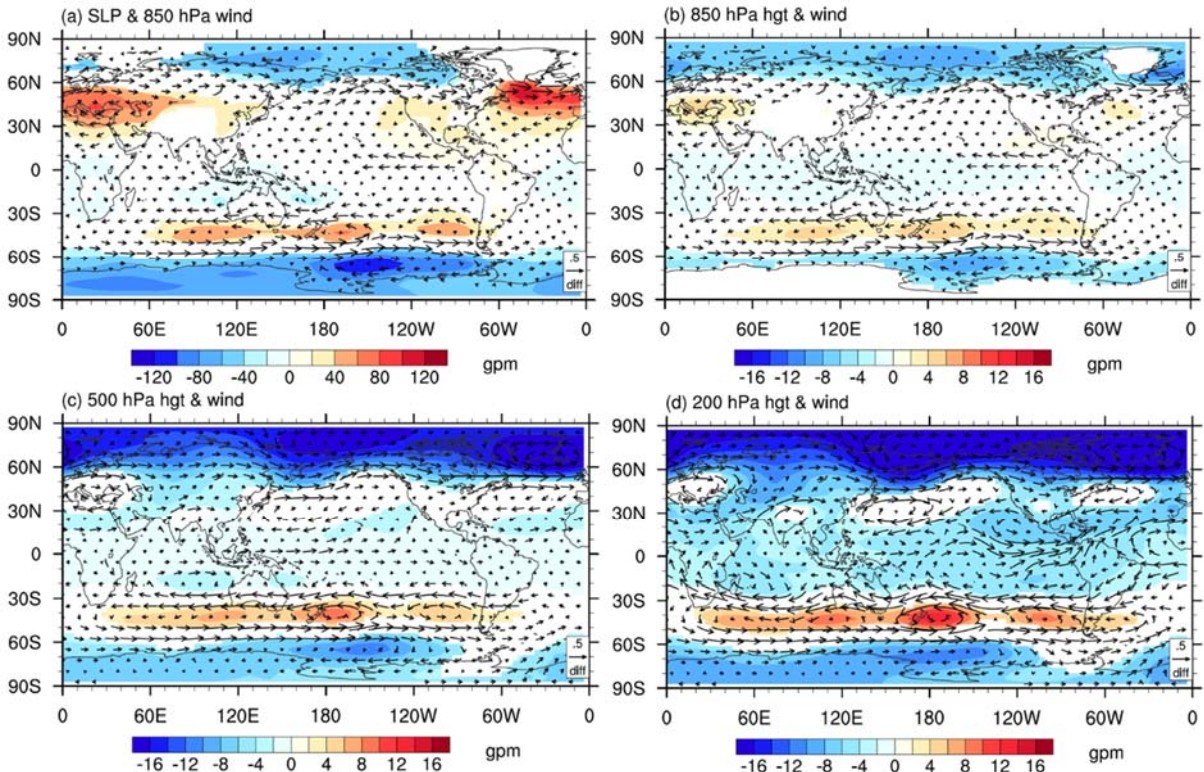


**Figure 5** Differences of annual mean (a) sea level pressure and 850 hPa wind, (b)

geopotential height and wind on 850 hPa, (c) geopotential height and wind on 500

hPa and (d) geopotential height and wind on 200 hPa between cold and warm periods

derived from the TraCE-ALL run. Those regions where significant above 95%

confidence level are plotted.

752

753

754

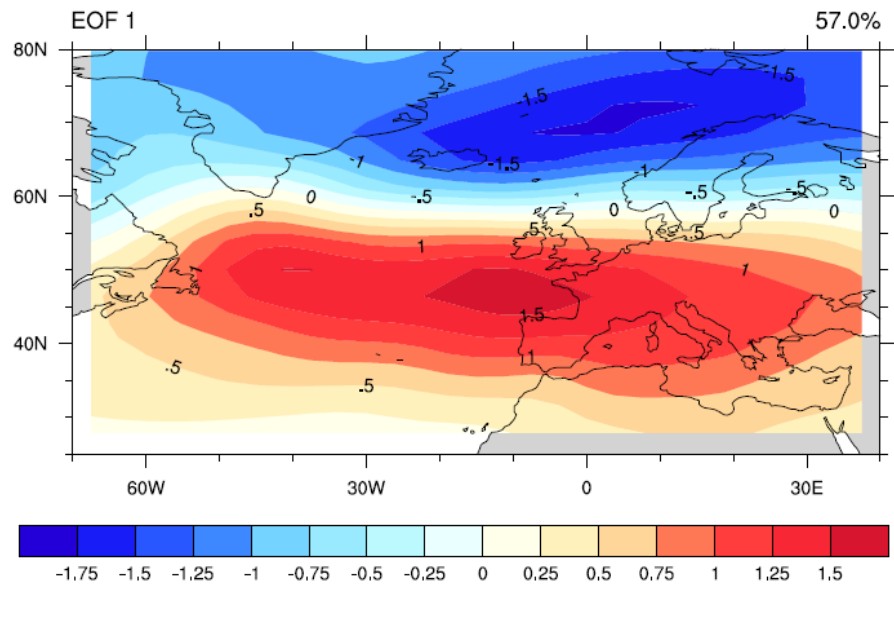

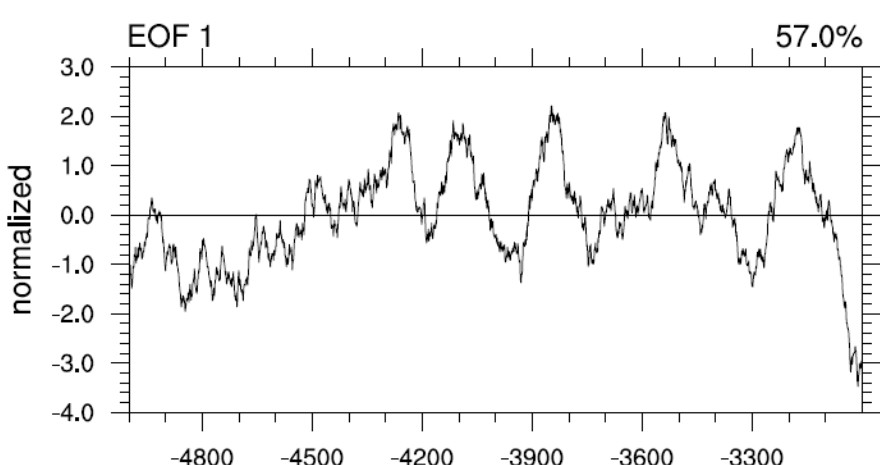

**Figure 6** Standardized first leading mode of the EOF of annual mean SLP over the
North Atlantic region (70W-40E, 25N-80N) during the period of 5.0 ka BP to 3.0 ka
BP derived from the TraCE-ALL run, after application of a 101-year running mean.
The spatial distribution is shown in the top panel, and the time series is shown in the
bottom panel. Only this mode passed the North test for EOF.

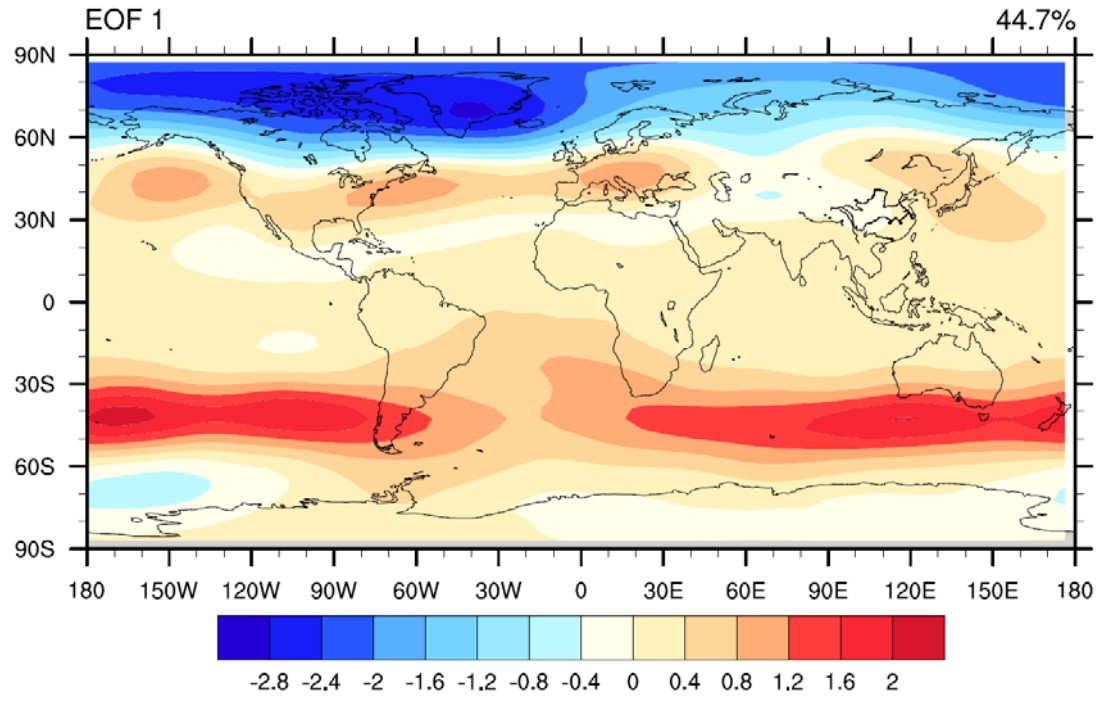

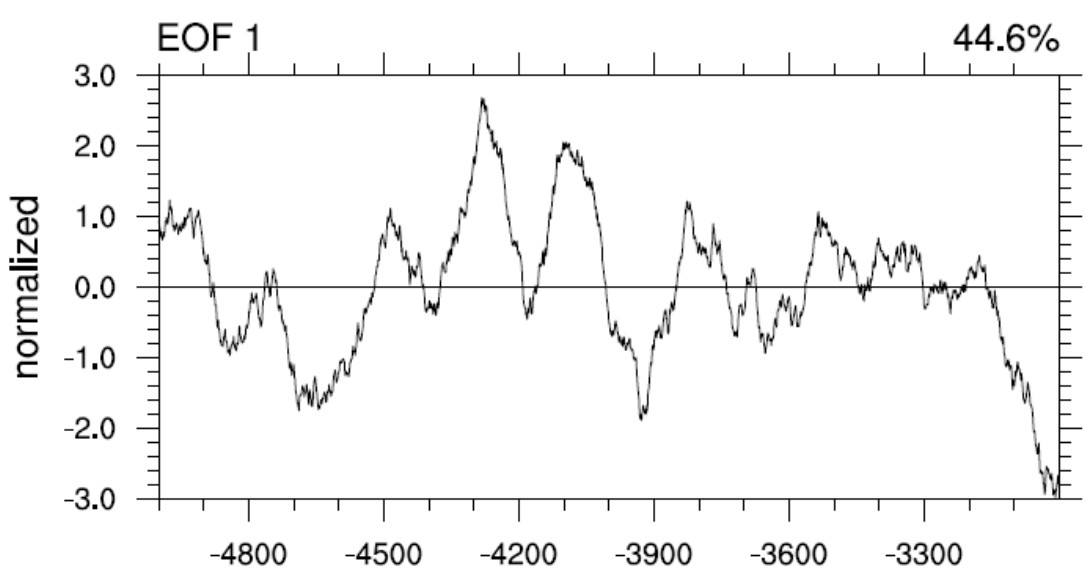

**Figure 7** Standardized first leading mode of the EOF of annual mean geopotential height at 200 hPa during the period of 5.0 ka BP to 3.0 ka BP derived from the TraCE-ALL run, after application of a 101-year running mean. The spatial distribution is shown in the top panel, and the time series is shown in the bottom panel. Only this mode passed the North test for EOF.

780

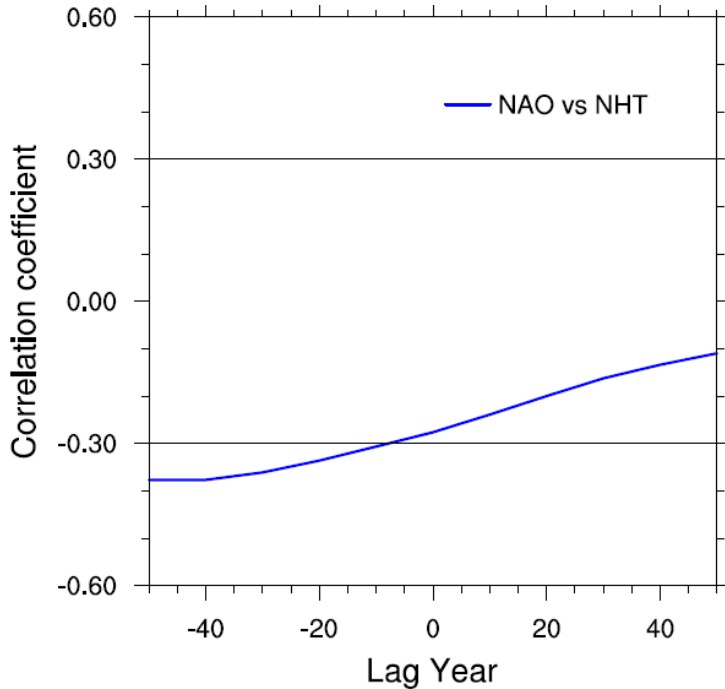

781

**Figure 8** Lead-lag correlation between the annual mean North Atlantic Oscillation

(NAO) and the North Hemisphere Surface Temperature (NHT) during 4.4 ka BP-4.0

ka BP derived from the TraCE-ALL run. The black lines (±0.3) show the significance

levels (p<0.05).


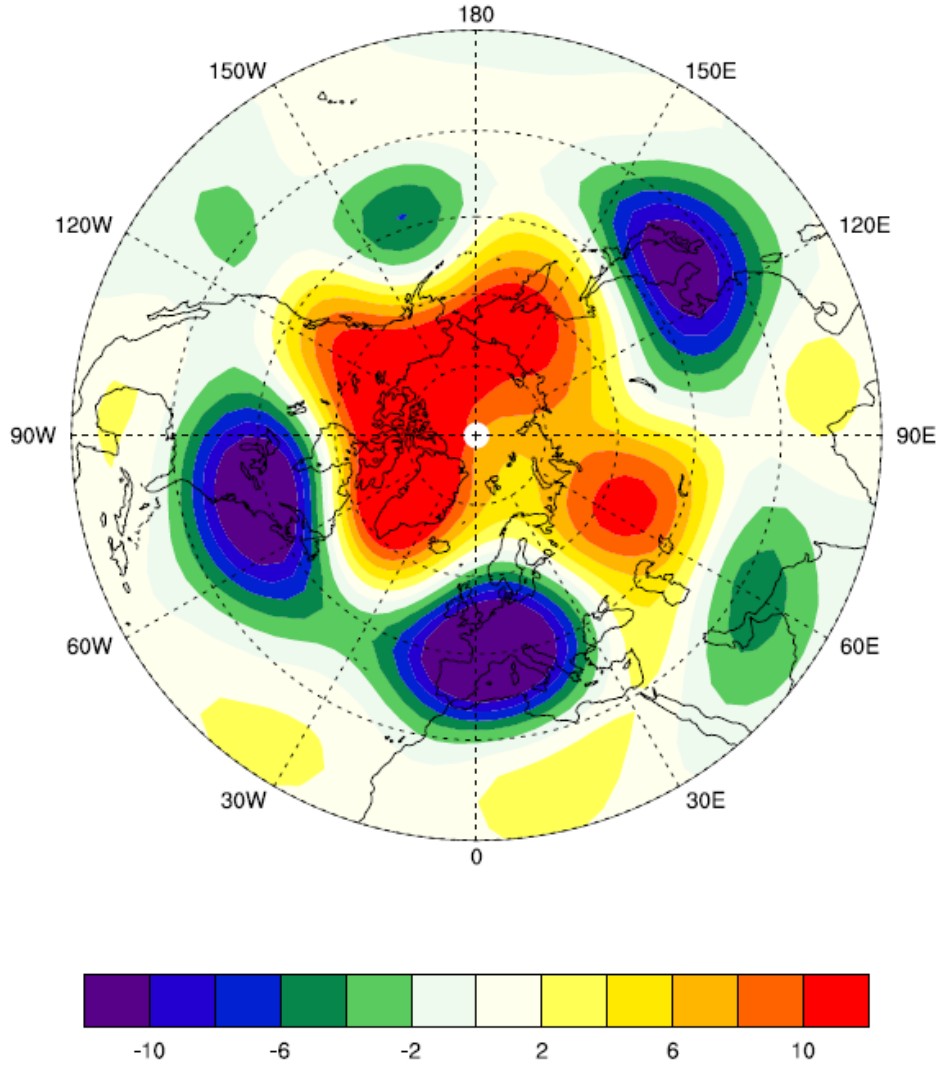


**Figure 9** Annual mean geopotential height regressed against the SST over the North

Atlantic during 5.0 ka BP - 3.0 ka BP derived from the TraCE-ALL run, after 31-year

running mean application.




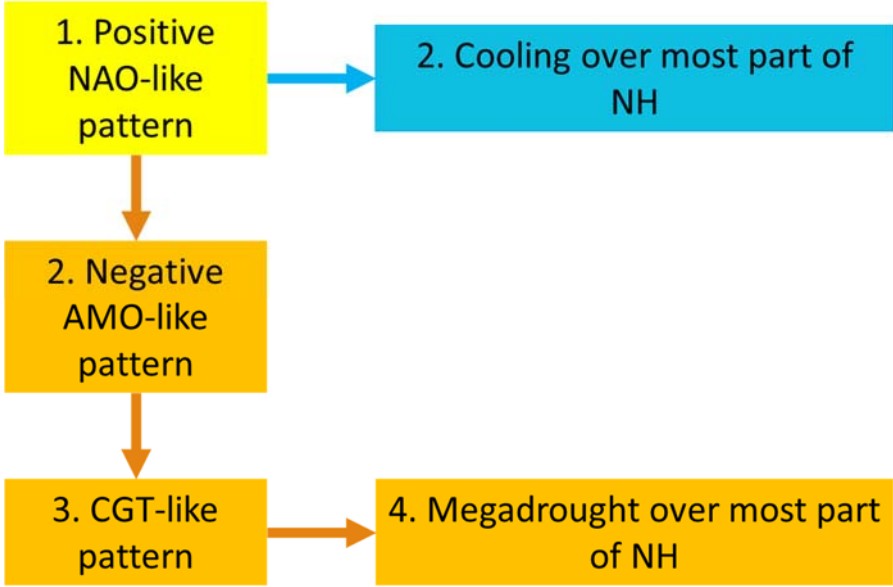


**Figure 10** Schematic diagram shown the mechanisms behind the 4.2 ka BP event.



**Table 1** The information of the experiments used in this study.

| Experiments | Forcings | Time spanning | Temporal resolution |
|---|---|---|---|
| **TraCE-ALL** | Orbital, melt-water flux, continental ice-sheet, and Greenhouse gases | 22000 BP to 1990 CE | Monthly mean |
| **TraCE-ORB** | Orbital only | 22000 BP to 1990 CE | Decadal mean |
| **TraCE-MWF** | Melt-water flux only | 19000 BP to 1990 CE | Decadal mean |
| **TraCE-ICE** | Continental ice-sheets only | 19000 BP to 1990 CE | Decadal mean |
| **TraCE-GHG** | Greenhouse gases only | 22000 BP to 1990 CE | Decadal mean |




**Table 2** Correlation coefficients between the annual mean and seasonal mean NHTs
derived from the TraCE-ALL run and those from each single-forcing run from 5.0 ka
BP to 3.0 ka BP.

| Single forcing run | Annual mean | JJA mean | DJF mean |
|:---:|:---:|:---:|:---:|
| TraCE-ORB | -0.05 | 0.79 | -0.12 |
| TraCE-MWF | -0.18 | 0.48 | -0.43 |
| TraCE-ICE | -0.30 | -0.20 | -0.18 |
| TraCE-GHG | 0.14 | -0.73 | 0.40 |

