# Peer review of "Physical processes of cooling and megadrought in 4.2 ka BP event: results from TraCE-21ka simulations"

_Climate of the Past, 2018_

## Referee Comment (RC1) · Anonymous Referee #1 · 1 Nov 2018

4.2K BP event is a hot topic issue. However, the cause of 4.2K BP event is remaining unclear. This study made contribution to understand how teh 4.2k BP event occurred, using a group simulation consisting of full forcing experiment and multpile single-forcing experiments. Through comparing the results from these experiments with each other, this study draws a conclusion that the 4.2K BP event is induced by internal variability. I recommend to accept this manuscript, but some issue should be addressed. 1. Line 193-194, warming over the SH could be related to the orbital change, which induces insolation increases over the SH but decreases over the NH. How to approve this result. I recommend to plot the temperautre anomaly spatial distribution induced by oribital forcing. 2. Line 202-203, Over East China, the precipitation anomalies show a wet

south dry north pattern. The figure 3b could not support this result, since the signal is too weak to be insignificant. 3. Line 223-224. The subtropical highs and the relative anticyclones in both the SH and NH are strengthened. We only find the strengthened subtropic high over SH while we could not find the strengthened subtropic highs over NH (Figure 5c). Please examine it carefully. 4. Line 235-236. with higher SLP over land and lower SLP over the adjacent ocean (Fig. 5a). We also could not find this character. Please examine it carefully. 5. Line 244-245. The land over the SH experiences cool but wet conditions, and the mid latitude SH ocean is warmer. Is there proxy-based evidence over SH? 6. Figure 6. Please clarify the spatial domain of the EOF.

---

## Referee Comment (RC2) · Anonymous Referee #2 · 2 Nov 2018

This paper uses a transient model simulation (TraCE-21ka) to explore the possible causes of the 4.2 ka event. While various hypotheses exist regarding the causes of this event, this remains an open and interesting question. The authors find evidence in the transient simulation for climate fluctuations in the middle Holocene that show some of the same temporal and spatial patterns as the 4.2 ka event, and through analysis using several additional single-forcing experiments, argue that the fluctuations likely arose through internal variability of the climate system. The results support some previous hypotheses and work on the causes of this event, and the paper does make a contribution in its use and analysis of the TraCE simulation.

[Figure]

I had several major concerns with the paper, including the overlap between this paper and another paper by the same authors that is under review in Climate of the Past, as well as how well some of the conclusions are supported by the results. These concerns are described in more detail below.

Major comments

1. The authors have another paper under review in Climate of the Past ("Comparing the spatial patterns of climate change in the 9th and 5th millennia B.P. from TRACE-21 model simulations" by Ning, Liu, Bradley and Yan) that has significant overlap with this manuscript. The Ning et al. paper uses the same model simulation (TraCE-21ka) to analyze the 8.2 ka and 4.2 ka events. Both papers present analysis using the same techniques (anomaly maps, principal component analysis). The Yan et al. paper (this review) provides a more in-depth analysis of the 4.2 ka event, but it is unclear why two papers are necessary. Perhaps even more important, the two papers come to conflicting conclusions about the cause of the 4.2 ka event. In Ning et al., it is stated "We speculate that long term changes in insolation related to precessional forcing led to cooling, which passed a threshold around 4500 years B.P., leading to a reduction in the AMOC and associated teleconnections across the globe. Based on widespread paleoclimatic evidence for the onset of neoglaciation (Solomina et al., 2015), it seems clear that there was a fundamental shift in climate around this time." Whereas, Yan et al. argue that stochastic variability internal to the climate system caused the 4.2 ka event independent of any external climate forcing.

2. It would useful to show on Figure 3 the locations of proxy records discussed in the text that document anomalies at 4.2 ka (perhaps circles color coded according to whether proxy anomalies were cold/warm or wet/dry during the event). This would help to make the point that the model event has the appropriate spatial pattern.

3. The authors need to discuss the implications that their maps show the difference between warm event and cold event – namely, that this approach amplifies the model

anomaly as compared to taking the difference between cold event and long-term average, which is probably what most of the proxy records are showing. Specifically, the authors should also discuss whether differences between cold event and long-term average (say averaged 500-1000 years before the event) are statistically significant. More generally, the authors need to make a point of discussing that the size of the modeled anomalies ARE VERY SMALL. I think it is fine to use the simulations to put forth a hypothesis about processes causing the 4.2 ka event, but given the small size of the modeled changes, it is also very important to be clear that we still might not be modeling events comparable to the 4.2 ka event (e.g., make this point clearer on lines 253-254).

4. Analysis of AMOC: The authors mention several times that simulated patterns are similar to those caused by AMOC, but AMOC is not analyzed. Further, the Ning et al. paper specifically attributes the event to AMOC changes. It is not difficult to generate an AMOC time series from TraCE (e.g., maximum of the meridional overturning streamfunction – the variable 'MOC' – over the North Atlantic avoiding the surface wind-mixed layer) and this would greatly help to clarify what the role of AMOC is.

5. Lines 289-292: The difference between the sum of the single-forcing experiments and the ALL simulation is not strictly internal variability. The difference will also include any interactions between the single forcings. This should be more clearly stated on these lines. Also on Line 295: add "in isolation" to the end of the phrase "the 4.2 ka event might not be triggered by those external forcings" because it is possible that interactions between forcings could be important.

6. Line 202-204, 234-239: Precipitation changes in China are largely insignificant. Recommend deleting these sentences.

Minor comments

Line 20: Change "many" to "several"

Line 46: "there were warming periods in Holocene induced by natural forcing compa­rable to current warming." Current warming, being driven by increased atmospheric greenhouse gas concentrations cannot by definition be comparable to any warming periods in the Holocene. Do you mean comparable in size? Even then, this is debat­able.

Line 57-58: "that inaugurated the "modern" El Nino Southern Oscillation (Fisher et al. 2008)." The record cited is not a direct record of ENSO (it is an ice core in the Yukon) and there are lots of more direct records of ENSO from the tropical Pacific that suggest complexity in how ENSO changed through the middle to late Holocene. Delete this phrase.

Line 70: "Moreover, according to the hydrologic cycle. . ." I'm not sure what this means. Is the point that hydroclimate changes are often regionally specific, and other regions could have had different hydroclimate changes?

Lines 76-80: "For the causes of the event, some reconstruction studies have suggested that orbital forcing played an important role in the early Holocene. . ." Does this refer to abrupt changes in the early Holocene, or longer-scale changes? Please provide references. ". . .; however, no strong evidence has shown that the solar forcing affected glacier fluctuations (cooling events) in the late Holocene. . ." Does "solar forcing" here refer to solar irradiance changes or to orbital forcing? Also, glacier fluctuations are only one indication of cooling, other temperature proxies do seem to be sensitive to solar irradiance changes.

Lines 90-91: For clarity, change "Additionally, there are discrepancies in the circulation pattern during the late Holocene (Finkenbinder et al., 2016)" to something like "How­ever, studies come to differing conclusions on the likely phase of the NAO-like pattern during the late Holocene."

Line 94: Change "might could be" to "could be"

Line 159: It is important to be very careful about calling a particular event in the model simulations the 4.2 ka event, especially since the variability being described in the model is internally driven. It is likely coincidental that these events described in the TraCE experiment happen around 4.2 ka – particularly if they are the result of internal variability. It is more appropriate to say "which indicates that simulated climate events potentially comparable to the 4.2 ka event" instead of "which indicates that the 4.2 ka BP event has multidecadal to centennial variabilities." Similarly, on the following lines, use "Moreover, the centennial warming periods right before and after the simulated cooling event indicate that this event might be included in a quasi-millennium variation" instead of "Moreover, the centennial warming periods right before and after the 4.2 ka BP event indicate that this event might be included in a quasi-millennium variation."

Figure captions: Specify which of the model simulations (i.e., "ALL") is plotted.

Figure 1: flip x axis so that time matches the sense of the x axis in Figure 2. Also, it seems that June insolation is not the most informative since the climate fluctuation in question is mostly a wintertime response and plots are all showing mean annual.

Figures 3, 4, 5, 7: Plot full globe, 90 degrees south to 90 degrees north.

Line 197: change "most regions" to "many land regions"

Line 198: change "central and southern North America (Intra America)" to "interior North America and central America"

Line 212-213: There are many more citations of relevance here, going back to Vellinga and Wood (2002) Climatic Change 54: 251-267 and Zhang and Delworth (2005) Journal of Climate 18: 1853.

Line 252: Change "The solar irradiance is not included…" To "Changes in solar irradiance are not included…" Solar irradiance is included in this model, it is just not changing.

Lines 273-276: Clarify here that there was no meltwater flux applied in the model for

the years analyzed (5000-3000 years BP). Why might these correlation coefficients be significant given that there is no meltwater flux? Is this likely due to chance? Please discuss this more in the paper.

Lines 368-369: "We attributed the internal variabilities to be an essential forcing of the 4.2 ka BP event; however, why it occurs at approximately 4400 BP to 4000 BP remains unknown." If the event is stochastic (as argued), there is nothing more to know about why it occurred when it did.

Acknowledgements: The TraCE-21ka team and funding should be acknowledged. See instructions at: https://www.earthsystemgrid.org/project/trace.html.

References: There are other papers that have hypothesized links between the North Atlantic and the 4.2 ka event and that should be cited. They include: Cullen, H. M., Kaplan, A., Arkin, P. A., and deMenocal, P. B.: Impact of the North Atlantic Oscillation on Middle Eastern climate and streamflow, Climatic Change, 55, 315–338, 2002. Kushnir, Y. and Stein, M.: North Atlantic influence on 19th–20th century rainfall in the Dead Sea watershed, teleconnections with the Sahel, and implication for the Holocene climate fluctuations, Quaternary Sci. Rev., 29, 3843–3860, 2010.

---

## Author Comment (AC1) · 29 Dec 2018

Thank you very much for the comments. We have modified the manuscript carefully according to your valuable comments. The point-to-point replies are as follows:

1. Line 193-194, warming over the SH could be related to the orbital change, which induces insolation increases over the SH but decreases over the NH. How to approve this result. I recommend to plot the temperature anomaly spatial distribution induced by orbital forcing.

Reply: Thank you for point out this. The requested figure has been added in the revised

version as Figure S3. The temperature difference between the cold and warm period induced by the orbital forcing is calculated by the average of period 4200 BP-3900 BP minus the average of period 4800 BP-4500 BP.

2. Line 202-203, Over East China, the precipitation anomalies show a wet south dry north pattern. The figure 3b could not support this result, since the signal is too weak to be insignificant.

Reply: Thank you for point out this issue. Yes, it's not so significant. So a statement has been added in the revised version. Lines 215-219.

3. Line 223-224. The subtropical highs and the relative anticyclones in both the SH and NH are strengthened. We only find the strengthened subtropical high over SH while we could not find the strengthened subtropical highs over NH (Figure 5c). Please examine it carefully.

Reply: Thank you for point out this issue. The strengthened subtropical highs can be represented not only by pressure but also by wind field. But yes, the significant pressure changes are different over NH and SH, with most significant at low level over the NH while at high level over the SH. We added a statement to illustrate this phenomenon more carefully. Lines 242-244.

4. Line 235-236. With higher SLP over land and lower SLP over the adjacent ocean (Fig. 5a). We also could not find this character. Please examine it carefully.

Reply: The SLP gets higher over land significantly while lower over ocean not significantly. Yes, additional statements have added in the revised version. Lines 258-260.

5. Line 244-245. The land over the SH experiences cool but wet conditions, and the mid latitude SH ocean is warmer. Is there proxy-based evidence over SH?

Reply: Unfortunately, we don't have significant evidences right now. But we can tell from the southward shift of the ITCZ position, which tends to locate over the warmer hemisphere.

6. Figure 6. Please clarify the spatial domain of the EOF.

Reply: The region for the EOF is (70W-40E, 25N-80N), which has been added in the figure caption of Fig. 6.

---

## Author Comment (AC2) · 29 Dec 2018

Thank you very much for your constructive comments which improve our work. We have modified our manuscript carefully according to your comments. The point-to-point replies are as following:

Major comments

1. The authors have another paper under review in Climate of the Past ("Comparing the spatial patterns of climate change in the 9th and 5th millennia B.P. from TRACE-21 model simulations" by Ning, Liu, Bradley and Yan) that has significant overlap with this

manuscript. The Ning et al. paper uses the same model simulation (TraCE-21ka) to analyze the 8.2 ka and 4.2 ka events. Both papers present analysis using the same techniques (anomaly maps, principal component analysis). The Yan et al. paper (this review) provides a more in-depth analysis of the 4.2 ka event, but it is unclear why two papers are necessary. Perhaps even more important, the two papers come to conflicting conclusions about the cause of the 4.2 ka event. In Ning et al., it is stated "We speculate that long term changes in insolation related to precessional forcing led to cooling, which passed a threshold around 4500 years B.P., leading to a reduction in the AMOC and associated teleconnections across the globe. Based on widespread paleoclimatic evidence for the onset of neoglaciation (Solomina et al., 2015), it seems clear that there was a fundamental shift in climate around this time." Whereas, Yan et al. argue that stochastic variability internal to the climate system caused the 4.2 ka event independent of any external climate forcing.

Reply: Thank you very much for your valuable comments. We agree with you that we need to clarify the difference between the two papers. This is a very important issue that we should address carefully. First, the two papers focus on different aspects about the 4.2 ka BP event. Ning et al. paper focuses on the spatial patterns, comparing the differences between the 9th and the 5th millennial BP using TraCE simulations on a long-term perspective, and concludes that there might be a phase transition of AMOC from stronger state in the beginning of the 5th millennial BP to weaker state through most of the late Holocene due to the reduction of orbital-insolation, but with fluctuations. Yan et al. also uses TraCE simulations but mainly focuses on the physical mechanisms and possible causes of the 4.2 ka BP event, and concludes that the fluctuations relating to the 4.2 ka BP event might be related to the internal variabilities. It is possible that the internal variability related fluctuations be superposed on the orbital related long-term trend found in Ning et al. paper. Therefore, the results of the two papers are not incompatible. Actually, we also mentioned the role of external forcings in Yan et al. paper, in the revised version, we make this point more clearly. Second, unlike the reconstruction work, different aspects can be investigated

based on the same set of climate experiments from different perspectives, e.g. from the atmospheric point of view, from the oceanic point of view, and from the time scales point of view etc, because the climate model is physical-based and spatiotemporally continuous. Especially from the time scales point of view, we may get different results, since the mechanisms and the causes may differ on different time scales. The results from Ning et al. paper is on a longer time scale than Yan et al. paper. So the two papers are actually complementary. Third, the EOF is an efficient method to illustrate the leading modes of climate changes. Ning et al. paper uses it to show the main patterns and the evolution of the SST during the two periods. While Yan et al. paper uses EOF to show the leading circulation change during the period of around 4.2 ka BP event. Before we analyze the physical mechanisms of the event, we need to make sure that the model performance is reasonable comparing to the records, which might be the concern of overlap between the two papers, but this is necessary. Therefore, to address your concerns, we have modified the manuscript to show the different aspects that our manuscript focuses on from Ning et al. paper.

2. It would useful to show on Figure 3 the locations of proxy records discussed in the text that document anomalies at 4.2 ka (perhaps circles color coded according to whether proxy anomalies were cold/warm or wet/dry during the event). This would help to make the point that the model event has the appropriate spatial pattern.

Reply: Yes, thank you for pointing out this issue. Some sites denoting the wet/dry anomalies during the event are added in the revised Fig. 3b, and the relative model-data comparison has been added in the revised version. Lines 206-209.

3. The authors need to discuss the implications that their maps show the difference between warm event and cold event – namely, that this approach amplifies the model anomaly as compared to taking the difference between cold event and long-term average, which is probably what most of the proxy records are showing. Specifically, the authors should also discuss whether differences between cold event and long-term average (say averaged 500-1000 years before the event) are statistically significant.

[Figure]

More generally, the authors need to make a point of discussing that the size of the modeled anomalies ARE VERY SMALL. I think it is fine to use the simulations to put forth a hypothesis about processes causing the 4.2 ka event, but given the small size of the modeled changes, it is also very important to be clear that we still might not be modeling events comparable to the 4.2 ka event (e.g., make this point clearer on lines 253-254).

Reply: Yes, the model responses to the external forcings are small even giving a strong forcing, especially in the Holocene because of the absence of a significant change of the AMOC and the meltwater forcing after 6 ka (Liu et al., 2014). So we use the amplified anomaly between the cold and warm periods to investigate the possible mechanisms. It is still significantly cool and dry over Europe and Central America, given the temperature and precipitation differences between the cold event and the long-term average (averaged over 5000-3000 BP) (Fig. A). And the North Atlantic region is still the active center. But yes, we need to clarify the smaller signal in the modeled changes. The statement of "but with smaller magnitude" has been added in the revised version, Line 276. The required statement has been added in the last section, Lines 430-437.

4. Analysis of AMOC: The authors mention several times that simulated patterns are similar to those caused by AMOC, but AMOC is not analyzed. Further, the Ning et al. paper specifically attributes the event to AMOC changes. It is not difficult to generate an AMOC time series from TraCE (e.g., maximum of the meridional overturning streamfunction – the variable 'MOC' – over the North Atlantic avoiding the surface wind-mixed layer) and this would greatly help to clarify what the role of AMOC is.

Reply: Thank you for pointing out this issue. Yes, we can easily provide the time series of the simulated AMOC (Fig. B). It is clear that the 8.2 ka BP event-related AMOC is dominated by the meltwater forcing, as has been revealed by Ning et al. paper. However, for the 4.2 ka BP event-related AMOC, it might be forced by both the orbital variation on longer time scale (orange line in Fig. B) and internal variation on centennial time scale. From this point of view, this paper and Ning et al. paper are actually not

incompatible. We just put forward these two possibilities. Figure B has been added in the revised version as Fig. S4, along with the spatial pattern of the temperature and precipitation differences between the weak and strong AMOC states. We have added statements about the AMOC in the discussion part, Lines 390-396.

5. Lines 289-292: The difference between the sum of the single-forcing experiments and the ALL simulation is not strictly internal variability. The difference will also include any interactions between the single forcings. This should be more clearly stated on these lines. Also on Line 295: add "in isolation" to the end of the phrase "the 4.2 ka event might not be triggered by those external forcings" because it is possible that interactions between forcings could be important.

Reply: Yes, you are right that the forcings and internal variability can be interacted between each other. We need to consider the linearity of climate responses to the external forcings at different spatial scales. Additional statements have been added in the revised version. Lines 318-325. Yes, "in isolation" has been added in the revised version. Line 329.

6. Line 202-204, 234-239: Precipitation changes in China are largely insignificant. Recommend deleting these sentences.

Reply: Although not so significant, we can still draw a general view on how the climate changes over China. But yes, we need further investigations with higher resolutions of modeling works and reconstruction works. The statements have been changed in the revised version. Lines 215-219, 258-260.

Minor comments Line 20: Change "many" to "several"

Reply: Changed, Lines 20-21.

Line 46: "there were warming periods in Holocene induced by natural forcing comparable to current warming." Current warming, being driven by increased atmospheric greenhouse gas concentrations cannot by definition be comparable to any warming

periods in the Holocene. Do you mean comparable in size? Even then, this is debatable.

Reply: There are some reconstructed records reported that there are some historical warming periods comparable to the current warming period in magnitude. But yes, this is debatable. We deleted this statement in the revised version. Lines 47-48.

Line 57-58: "that inaugurated the "modern" El Nino Southern Oscillation (Fisher et al. 2008)." The record cited is not a direct record of ENSO (it is an ice core in the Yukon) and there are lots of more direct records of ENSO from the tropical Pacific that suggest complexity in how ENSO changed through the middle to late Holocene. Delete this phrase.

Reply: Yes, deleted. Lines 59-60.

Line 70: "Moreover, according to the hydrologic cycle: : :" I'm not sure what this means. Is the point that hydroclimate changes are often regionally specific, and other regions could have had different hydroclimate changes?

Reply: Yes, from the perspective of hydrology balance, the hydroclimate changes are often regionally specific. This has been added in the revised version. Lines 72-73.

Lines 76-80: "For the causes of the event, some reconstruction studies have suggested that orbital forcing played an important role in the early Holocene: : :" Does this refer to abrupt changes in the early Holocene, or longer-scale changes? Please provide references. ": : :; however, no strong evidence has shown that the solar forcing affected glacier fluctuations (cooling events) in the late Holocene: : :" Does "solar forcing" here refer to solar irradiance changes or to orbital forcing? Also, glacier fluctuations are only one indication of cooling, other temperature proxies do seem to be sensitive to solar irradiance changes.

Reply: "For the causes of the event, some reconstruction studies have suggested that orbital forcing played an important role in the early Holocene: : :", we mean the abrupt

changes in the early Holocene. The "solar forcing" refers to the solar irradiance change. But yes, I understand what you mean. We should use the same forcing here. So we have changed the statement and clarified the forcing to be "solar irradiance". The references (Wang et al., 2005; Rupper et al., 2009; Owen and Dortch, 2014) are provided in the revised version. Lines 79-85.

Lines 90-91: For clarity, change "Additionally, there are discrepancies in the circulation pattern during the late Holocene (Finkenbinder et al., 2016)" to something like "However, studies come to differing conclusions on the likely phase of the NAO-like pattern during the late Holocene."

Reply: Yes, changed. Lines 95-97.

Line 94: Change "might could be" to "could be"

Reply: Changed. Line 100.

Line 159: It is important to be very careful about calling a particular event in the model simulations the 4.2 ka event, especially since the variability being described in the model is internally driven. It is likely coincidental that these events described in the TraCE experiment happen around 4.2 ka – particularly if they are the result of internal variability. It is more appropriate to say "which indicates that simulated climate events potentially comparable to the 4.2 ka event" instead of "which indicates that the 4.2 ka BP event has multidecadal to centennial variabilities." Similarly, on the following lines, use "Moreover, the centennial warming periods right before and after the simulated cooling event indicate that this event might be included in a quasi-millennium variation" instead of "Moreover, the centennial warming periods right before and after the 4.2 ka BP event indicate that this event might be included in a quasi-millennium variation."

Reply: Yes, you are right. Thank you for pointing out this. It would be coincidental that the cooling events in the TraCE run happen around 4.2 ka if they ARE induced by the internal variability. The statements are changed in the revised version. Lines 167-169.

Figure captions: Specify which of the model simulations (i.e., "ALL") is plotted.

Reply: Yes, the statements of "derived from the TraCE-ALL run" have been added into the figure captions of Figures 3-9.

Figure 1: flip x axis so that time matches the sense of the x axis in Figure 2. Also, it seems that June insolation is not the most informative since the climate fluctuation in question is mostly a wintertime response and plots are all showing mean annual.

Reply: Fig. 1 has been modified and limited to the period of 5000-3000 BP. Yes, the orbital-induced insolation changes are seasonal dependent. Here we use summer insolation as an example, since the long-term variations in Northern Hemisphere summer insolation are generally thought to control glaciation (Huybers, 2006). Also, according to the Milankovitch theory, the ice age cycles are paced by the Earth's orbital variations, with Northern Hemisphere summer insolation intensity playing a dominant role in the growth and decay of Northern Hemisphere ice sheets (TraCE-21ka Description, http://www.cgd.ucar.edu/ccr/TraCE/).

Figures 3, 4, 5, 7: Plot full globe, 90 degrees south to 90 degrees north.

Reply: The figures have been re-plotted with a full global map.

Line 197: change "most regions" to "many land regions"

Reply: Changed. Line 204.

Line 198: change "central and southern North America (Intra America)" to "interior North America and central America"

Reply: Changed. Lines 205-206.

Line 212-213: There are many more citations of relevance here, going back to Vellinga and Wood (2002) Climatic Change 54: 251-267 and Zhang and Delworth (2005) Journal of Climate 18: 1853.

Reply: Yes, thank you for providing the references. The references have been added in the revised version, and Figure S4 is added to illustrate the spatial pattern of the temperature and precipitation differences between the weak and strong AMOC states. Lines 228-229.

Line 252: Change "The solar irradiance is not included: : :" To "Changes in solar irradiance are not included: : :" Solar irradiance is included in this model, it is just not changing.

Reply: Yes, you're right. Changed in the revised version. Lines 273-274.

Lines 273-276: Clarify here that there was no meltwater flux applied in the model for the years analyzed (5000-3000 years BP). Why might these correlation coefficients be significant given that there is no meltwater flux? Is this likely due to chance? Please discuss this more in the paper.

Reply: Yes, thank you for pointing out this issue. A short discussion has been added in the revised version. Lines 298-304.

Lines 368-369: "We attributed the internal variabilities to be an essential forcing of the 4.2 ka BP event; however, why it occurs at approximately 4400 BP to 4000 BP remains unknown." If the event is stochastic (as argued), there is nothing more to know about why it occurred when it did.

Reply: Yes, it would be stochastic if it IS forced by the internal variability. However, we still need more evidences and modeling works to make sure that this event IS forced by the internal variability. We have changed the statement in the revised version. Lines 419-423.

Acknowledgements: The TraCE-21ka team and funding should be acknowledged. See instructions at: https://www.earthsystemgrid.org/project/trace.html.

Reply: Thank you for pointing out this, the acknowledgement has been added in the revised version. Lines 447-449.

References: There are other papers that have hypothesized links between the North Atlantic and the 4.2 ka event and that should be cited. They include: Cullen, H. M., Kaplan, A., Arkin, P. A., and deMenocal, P. B.: Impact of the North Atlantic Oscillation on Middle Eastern climate and streamflow, Climatic Change, 55, 315–338, 2002. Kushnir, Y. and Stein, M.: North Atlantic influence on 19th–20th century rainfall in the Dead Sea watershed, teleconnections with the Sahel, and implication for the Holocene climate fluctuations, Quaternary Sci. Rev., 29, 3843–3860, 2010.

Reply: Thank you for providing the references. The mentioned and additional references have been added in the revised version. Lines 352-354, 384-389.

References: Booth, R. K., Jackson, S. T., Forman, S. L., Kutzbach, J. E., Bettis, I. E. A., Kreig, J., and Wright, D. K.: A severe centennial-scale drought in mid-continental North America 4200 years ago and apparent global linkages, The Holocene, 15, 321-328, 10.1191/0959683605hl825ft, 2005.

Huybers, P.: Early Pleistocene glacial cycles and the integrated summer insolation forcing, Science, 313, 508-511, 10.1126/science.1125249, 2006.

Kim, J.-H., Rimbu, N., Lorenz, S. J., Lohmann, G., Nam, S.-I., Schouten, S., Rühlemann, C., and Schneider, R. R.: North Pacific and North Atlantic sea-surface temperature variability during the Holocene, Quaternary Science Reviews, 23, 2141-2154, 10.1016/j.quascirev.2004.08.010, 2004.

Liu, Z., Zhu, J., Rosenthal, Y., Zhang, X., Otto-Bliesner, B. L., Timmermann, A., Smith, R. S., Lohmann, G., Zheng, W., and Elison Timm, O.: The Holocene temperature conundrum, Proceedings of the National Academy of Sciences of the United States of America, 111, E3501-3505, 10.1073/pnas.1407229111, 2014.

Marchant, R., and Hooghiemstra, H.: Rapid environmental change in African and South American tropics around 4000 years before present: a review, Earth-Science Reviews, 66, 217-260, 10.1016/j.earscirev.2004.01.003, 2004.

Owen, L. A., and Dortch, J. M.: Nature and timing of Quaternary glaciation in the Himalayan–Tibetan orogen, Quaternary Science Reviews, 88, 14-54, 10.1016/j.quascirev.2013.11.016, 2014.

Rupper, S., Roe, G., and Gillespie, A.: Spatial patterns of Holocene glacier advance and retreat in Central Asia, Quaternary Research, 72, 337-346, 10.1016/j.yqres.2009.03.007, 2009.

Wang, Y. J., Cheng, H., Edwards, L. R., He, Y. Q., Kong, X. G., An, Z. S., Wu, J. Y., Kelly, M., Dykoski, C. A., and Li, X. D.: The Holocene Asian Monsoon: Links to Solar Changes and North Atlantic Climate, Science, 308, 854-857, 2005.

Please refer to the supplementary for your convinience.

Please also note the supplement to this comment:
https://www.clim-past-discuss.net/cp-2018-131/cp-2018-131-AC2-supplement.pdf

———————————————————

[Figure]

[Figure]

Figure A Spatial distribution of the annual mean (a) surface temperature and (b) precipitation anomalies of the cold periods against the long-term average of 5000-3000 BP derived from the TraCE-ALL run. Those regions where significant above 95% confidence level are dotted.

**Fig. 1.** Figure A Spatial distribution of the annual mean (a) surface temperature and (b) precipitation anomalies of the cold periods against the long-term average of 5000-3000 BP

[Figure]

Figure B Time series of simulated AMOC over the past 22 ka
derived from the all forcing run (black dashed line) and each
single forcing run (colored lines). A 101-year running mean is
applied to the time series.

**Fig. 2.** Figure B Time series of simulated AMOC over the past 22 ka derived from the all forcing run (black dashed line) and each single forcing run (colored lines).

---

## Author Response (AR2)

Editor Decision: Publish subject to minor revisions (review by editor) (04 Feb 2019) by Monica Bini
Comments to the Author:
The authors made a good job in the review of the manuscript.
Few minor revisions are still required before the acceptance for publication:
Line 403: please, in order to be consistent with the text and the special issue, change "4400-4000" whit "4.4-4.0 ka BP"
Line 741 "Gulf of Omen" is "Gulf of Oman"
After these changes the manuscript will be accepted.

Reply: Thank you very much for your affirmation and comments.
We have changed "4400-4000" with "4.4-4.0 ka BP", and changed "Gulf of Omen" with "Gulf of Oman". Line 402 and Line 735.

[revised manuscript text omitted]

**Figure S4 (a)** Simulated AMOC over the past 22 ka derived from TraCE-ALL (black line) and each single forcing run (colored lines). Spatial distribution of the annual mean surface temperature difference (b) and precipitation difference (c) between the weak and strong AMOC states derived from TraCE-ALL. Those regions where significant above 95% confidence level are dotted in (b) and (c).
The weak and strong AMOC states are selected based on the standardized time series of AMOC during the period of past 6 ka, when the meltwater forcing is absent.

[Figure]

**Figure S5** Time series of annual mean NHT anomaly from 5000 BP to 3000 BP. Dashed black line is the result derived from the all forcing run. Red line indicates the difference between the result derived from all forcing run and that derived from the linear sum of the 4 single forcing runs.

[Figure]

**Figure S6** Standardized time series of the annual mean NHT derived from the TraCE-MWF run (blue line) and TraCE-ICE run (green line) from 5000 BP to 1990 CE. A 101-year running mean has been applied to the time series.

[Figure]

**Figure S7** Annual mean TS regressed against the NAO index leading 40-year during 4.4 ka BP - 4.0 ka BP.